# Effects of *Piper sarmentosum* on Metabolic Syndrome and Its Related Complications: A Review of Preclinical Evidence

Sophia Ogechi Ekeuku [ID], Mohd Fahami Nur Azlina [ID] and Kok-Yong Chin *[ID]

Department of Pharmacology, Faculty of Medicine, Universiti Kebangsaan Malaysia, Cheras,
Kuala Lumpur 56000, Malaysia; virgosapphire2088@yahoo.com (S.O.E.); nurazlinamf@ukm.edu.my (M.F.N.A.)
* Correspondence: chinkokyong@ppukm.ukm.edu.my; Tel.: +60-3-9145-9573

**Featured Application: This review illustrates the effects of *Piper sarmentosum* extract in the management of metabolic derangements constituting metabolic syndrome and their complications and suggests future direction to advance this field.**

**Abstract:** *Piper sarmentosum* (PS) is a traditional medicinal herb used by South East Asians. It demonstrates promising properties against various non-communicable diseases and infectious agents due to its antioxidant and anti-inflammatory properties. Given that oxidative stress and inflammation are involved in developing and exacerbating metabolic syndrome (MetS) and its principal components (central obesity, hyperglycaemia, hypertension, and dyslipidaemia), PS could manage MetS and its complications. This review summarises the available literature on the effects of PS on principal components of MetS and their complications. The accumulated evidence suggests that PS prevented adiposity, hyperglycaemia, hypertension, and dyslipidaemia in preclinical studies mainly through its antioxidant and anti-inflammatory properties. It also protected against MetS-associated cardiovascular complications. This review has identified research gaps in this field and suggested future studies to guide interested researchers to explore further or affirm the therapeutic potential of PS. One of the most significant challenges to the medical use of PS is the absence of randomised controlled trials in humans. This study gap must be bridged before PS supplementation could be used to manage MetS in humans.

**Keywords:** cholesterol; inflammation; oxidative stress; hypertension; obesity; diabetes

## 1. Introduction

Metabolic syndrome (MetS) is a multifactorial condition that develops due to the accumulation and chronification of several risk factors, including central obesity, insulin resistance, systemic hypertension, and dyslipidaemia [1,2]. MetS increases the risk of type 2 diabetes mellitus (T2DM), cardiovascular disease, and stroke, all of which impose significant morbidity and mortality worldwide [3,4]. A systematic review in 2016 showed that 12–37% of the Asian population and 12–26% of the European population suffered from MetS [5].

Calorie excess and physical inactivity are the main drivers of MetS. Elevated levels of circulating proinflammatory cytokines [6,7], oxidative stress markers, and low levels of antioxidants have been reported in MetS patients [8–11], underscoring their role in the pathogenesis of MetS. The excess calories consumed are stored as visceral adipose tissue, which actively secretes free fatty acid, proinflammatory cytokines, and adipokines, contributing to metabolic derangements [12]. In addition, nicotinamide adenine dinucleotide phosphate oxidase (NOX) is stimulated by the excess angiotensin II, resulting in oxidative stress due to increased radical oxygen species (ROS) production [13–16]. The combination of hormonal changes, chronic inflammation, and oxidative stress is harmful to various organ systems, particularly the cardiovascular system. Thus, an excess risk of cardiovascular diseases has been reported among patients with MetS [12].

Lifestyle and dietary modifications, as well as pharmacological interventions targeting energy metabolism pathways, are the current management strategy for MetS. However, lifestyle and dietary interventions require the long-term compliance of the patients to take effect. Furthermore, since patients often suffer from various metabolic conditions concurrently, each requires specific medications, issues of polypharmacy, drug–drug interactions, and increased medical costs are common [17]. Therefore, the pursuit of a pleiotropic agent to assist MetS management is ongoing. Medicinal plants are a potential source of agents that can fulfil this purpose. They contain various biologically active compounds with health benefits, consumed by local populations for generations [18–20]. *Piper sarmentosum* (PS) Roxb. is a herbaceous plant commonly found in South East Asia. It has been used traditionally to treat urological, dermatological, hepatobiliary, and gastric diseases. Apart from that, it is also known to possess antipyretic and anti-inflammatory properties [21]. Modern pharmacological studies have validated some medicinal properties of PS, such as antineoplastic [22], hypoglycaemic [23], and hypotensive [24] activities.

The biological effects of PS could be attributed to its anti-inflammatory activities [25–27]. Ethyl acetate and methanolic extracts from the leaves of PS were reported to reduce interleukin (IL)-1$\beta$ and tumour necrosis factor (TNF)-$\alpha$ expressions in murine microglial cells stimulated with $\beta$-amyloid [27]. In interferon-gamma/lipopolysaccharide-treated macrophages, PS methanolic extract showed significant nitric oxide (NO) inhibitory activity in a concentration-dependent manner, implying potential anti-inflammatory activities [25]. Animal studies also showed that ethanolic extract of PS root reduced acute and chronic inflammation by inhibiting ethyl phenylpropionate-induced ear oedema, carrageenan-induced paw oedema, and cotton pellet-induced granuloma formation in male rats [26]. Gas chromatography–mass spectrometry analysis of essential oils from leaves, fruits and roots of PS revealed the presence of myristicin and $\beta$-caryophyllene [28]. Myristicin decreased inflammation in polyinosinic-polycytidylic acid-induced RAW 264.7 cells by suppressing NO, cytokine, and chemokine production [29]. $\beta$-caryophyllene reduced hypoxia-induced inflammation in BV2 microglia cells by inhibiting the release of proinflammatory cytokines [30].

PS also possesses prominent antioxidant properties [31–34]. Aqueous (IC$_{50}$: 50.56 mg/mL) and ethanolic leaf extracts of PS (IC$_{50}$: 35.18 μg/mL) showed free radical scavenging activity via a 2,2-diphenyl-1-picrylhydrazyl assay [31,32]. Hafizah et al. [34] also reported that a PS crude extract displayed ferric-reducing power with the total phenolics content comparable to the Vitamin C standard. Aqueous, methanol, and hexane leaf extracts of PS prevented an increase in the malondialdehyde (MDA) level and activation of adaptive antioxidant defence in hydrogen peroxide ($H_2O_2$)-induced oxidative stress in human umbilical vein endothelial cells (HUVEC) [34]. The methanolic leaf extract of PS also reduced the MDA levels in the gastric tissue of male rats with stress-induced gastric ulcer [33]. Liquid chromatography–mass spectrometry analysis has revealed that the methanolic extract of PS contains several phytochemical constituents, including naringenin, quercetin, and hesperidin [35–37], which contribute to the overall antioxidant effects of PS.

This review aims to examine the current knowledge on the effects of PS on MetS and its related complications, given the rationale that the PS could potentially halt chronic inflammation and oxidative stress in MetS. The therapeutic effects of PS on each MetS component or their complications were discussed in the following sections. The discourse is followed by the authors' perspectives on the research gaps and future directions of this field. Finally, we hope the review can guide interested researchers to explore further or affirm the therapeutic potential of PS.

## 2. Literature Search

Relevant literature on this topic was gathered during August 2021 based on the search string "*Piper sarmentosum*" AND (adipose OR fat OR obesity OR "blood pressure" OR hypertension OR "blood glucose" OR hyperglycaemia OR insulin OR diabetes OR

cholesterol OR triglycerides OR "metabolic syndrome") using four electronic databases, i.e., PubMed, Scopus, Google Scholar, and Web of Science. Only original research articles written in English, with the primary objective of investigating the effects of PS on MetS, its principal components or its complications, were included in the review. Both clinical and preclinical studies were considered. The literature search yielded 438 unique articles, of which 22 papers were critically reviewed. The literature search process is summarised in Figure 1.

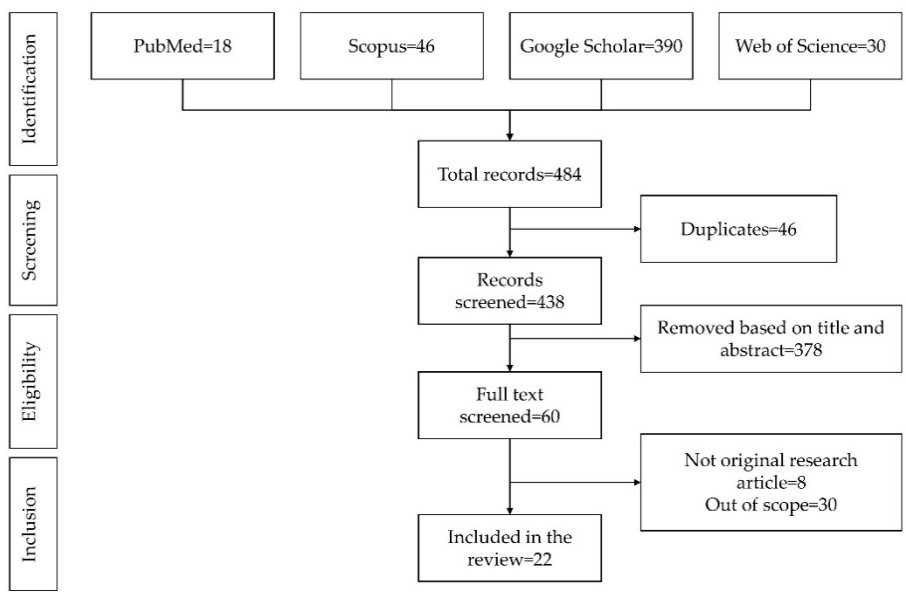

**Figure 1.** PRISMA flow chart for the literature search.

## 3. Effect of PS on Components of MetS

MetS is the culmination of several medical conditions, i.e., obesity, hyperglycaemia, hypertension, and dyslipidaemia, with interlacing underlying pathogenic pathways [1,2]. PS has been shown to exert protective effects against each condition in preclinical studies. To facilitate the discourse, the effects of PS on each condition and recommendations for future research are presented in the following sections.

### 3.1. Effect of PS on Obesity

Obesity is a major risk factor for developing MetS. Central obesity occurs when the excess energy in the body is stored in adipocytes, causing them to undergo hypertrophy [38]. The hypertrophied adipocytes actively secrete proinflammatory cytokines, such as IL-1, IL-6, and TNF-$\alpha$, resulting in low-grade chronic systemic inflammation [39,40]. Inadequate blood supply of the expanded adipose tissue results in cellular hypoxia and necrosis. Inflammation infiltration at the necrotic sites and the removal of cellular debris through phagocytosis could increase oxidative stress due to the release of free radicals such as NO and $H_2O_2$ [41,42], which further exacerbates the metabolic derangements [43]. Obesity is also associated with increased levels of the highly reactive advanced glycation and lipoxidation end-products [44,45], which support the development of other MetS components. Weight loss interventions, such as lifestyle, dietary, pharmacological, and surgical interventions, are frequently prescribed to MetS patients [12,46].

PS has been reported to reduce visceral fat in vivo [47]. Visceral fat is a hormonally active component of total body fat with distinct biochemical properties that influence various physiological and pathological processes in the human body [48]. Medical conditions such as metabolic syndrome and cardiovascular disease have been linked to abnormally high visceral fat deposition [48]. Chronic glucocorticoid therapy causes weight gain and visceral adiposity [49]. Excess glucocorticoids can cause a rise in circulating free fatty acids, lipid build-up in skeletal muscle and the liver, and adipocyte differentiation and

size, thereby increasing visceral adiposity [50]. In a study by Fairus et al. [47], PS aqueous extract (125 mg/kg/day for 48 days) reduced visceral fat deposition and the diameter of an adipocyte membrane in adrenalectomized male Sprague Dawley rats with obesity induced by dexamethasone. This suggests that PS might reduce lipid content in adipose tissues as the thickening of the adipocyte membrane is associated with an increased lipid content of adipose tissue [51]. However, the area, perimeter, and width of the individual perirenal adipocytes were not significantly different between the PS-treated and untreated groups. These findings suggested that it did not suppress adipocyte hypertrophy [47].

11β-hydroxysteroid dehydrogenase type 1 (11βHSD-1) is an enzyme that catalyses the conversion of 11-dehydrocorticosterone to corticosterone and vice versa, leading to hypertrophy and hyperplasia of adipocytes [47] and decreased adiponectin levels [52]. 11βHSD-1 activity has been reported to be increased in animal models of obesity and the subcutaneous adipose tissue of obese human subjects [53]. Adiponectin is an adipocyte-derived hormone negatively regulated by visceral fat accumulation [54]. Even though adiponectin is secreted by fat cells, obese people have significantly lower levels of adiponectin than non-obese people [55–57]. Women are reported to have higher plasma adiponectin levels than men, despite having a higher body fat percentage. According to Kern et al. [58], plasma adiponectin levels in women are 65% higher than in men, especially in relatively lean individuals. This sex difference in adiponectin levels has been confirmed by other studies [56,59].

Moreover, overexpression of 11βHSD-1 has been reported to increase leptin levels [60,61]. Leptin is a hormone secreted by adipocytes that reduces food intake and regulates body fat by increasing energy expenditure [62]. Leptin increases with adiposity [63,64], as evidenced by obese rodents and humans having higher levels of circulating leptin. These observations suggest the development of leptin resistance [64,65]. Leptin resistance promotes hunger and increases food intake, resulting in weight gain [66]. The actions of PS on adiponectin and leptin suggest it can influence satiety and adipose accumulation in the body.

PS aqueous leaves extract (1.25 mg/kg/day for 5 months) reduced the activity of 11βHSD-1 in the liver and adipose tissue of ovariectomy-induced obese female rats [53]. This observation suggests that PS supplementation prevents obesity by inhibiting 11βHSD-1 expression and activity, subsequently reducing adipocyte hypertrophy. The aqueous and methanolic leaf extracts of PS increased the circulating adiponectin levels in rats with ovariectomy-induced obesity [67] and fructose-induced MetS rats [68], suggesting that PS could prevent MetS and its components through up-regulating adiponectin levels. The methanolic leaf extract of PS reduced leptin levels in fructose-induced MetS rats [68]. Concurrently, food, fluid, and calorie intakes were reduced in MetS rats treated with methanolic extract of PS leaf, leading to a reduction in body adiposity and body weight [68]. These observations implied that PS supplementation reduced leptin resistance in MetS rats, which subsequently inhibited hunger, leading to a decreased fat mass and body weight. Meanwhile, the aqueous leaf extract of PS did not change the body weight in ovariectomy-induced obese rats [53,67].

Leptin resistance is also associated with increased oxidative stress [69]. Cell culture and animal studies demonstrated that oxidative stress increases adipocyte differentiation, hyperplasia, and hypertrophy, potentially leading to obesity [70–72]. Kumar et al. [68] suggested that the decrease in adipocyte size in fructose-induced MetS rats could be due to the antioxidant effects of active compounds in PS. However, this speculation has yet to be tested by any studies.

Overall, the current evidence suggests that PS supplementation inhibits obesity by regulating adipokine and leptin levels. Notably, leptin resistance was prevented by PS supplementation, leading to the suppression of food and calorie intake [68]. The reduction in calorie intake prevents adipocyte hypertrophy, increased body weight, and adiposity [68]. On the other hand, PS could suppress 11βHSD-1 activity and oxidative stress, leading to reduced adiposity [68].

To bridge the research gaps of this topic, the authors suggest combining calorie restriction and PS supplementation in future studies to assess the combinatorial interventions on obesity. This is because calorie restriction was shown to be the most effective dietary intervention for weight loss in humans [73]. Since the methanolic PS leaf extract effectively decreased body weight and calorie intake in fructose-induced MetS rats via regulating leptin and adiponectin levels [68], the combination could speed up weight loss in obese subjects. Furthermore, oxidative stress has been linked with increased adipocyte differentiation (adipogenesis). Further studies should examine adipogenesis markers, such as CCAAT/enhancer binding protein (CEBP)-α, -β, and peroxisome proliferator activator receptor (PPAR)-γ, to establish the effect of PS supplementation on adipocytes formation. Figure 2 summarises the current knowledge on the effect of PS supplementation on obesity and the further investigation required in this area.

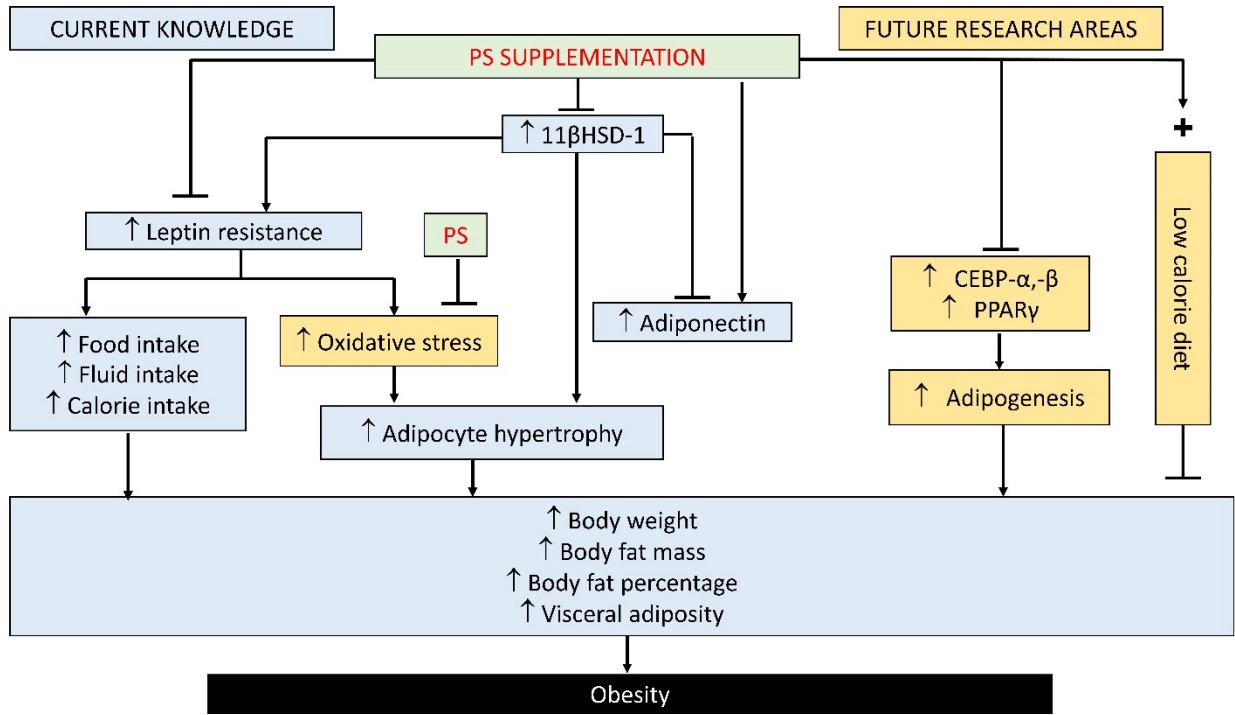

**Figure 2.** Protective effect of PS supplementation against obesity: current knowledge and future area of investigation. List of abbreviations: PS, *Piper sarmentosum*; 11βHSD-1, 11β-hydroxysteroid dehydrogenase type 1; CEBP-α, -β, CCAAT/enhancer-binding protein; PPAR-γ, peroxisome proliferator activator receptor-gamma; ↑, increase; +, in combination with.

### 3.2. Effect of PS on Hyperglycaemia or Diabetes

Populations with MetS are five times more likely to develop type 2 diabetes (T2DM) [74,75]. The main abnormality that connects the metabolic and hemodynamic disturbances found in MetS is impaired insulin-mediated glucose uptake [76,77]. Insulin resistance is linked to prediabetes, defined as an elevated plasma glucose level (hyperglycaemia) above the normal range but below the clinical diabetes threshold [78]. Hyperglycaemia in prediabetes can cause oxidative stress and upregulation of proinflammatory factors, leading to impaired insulin receptor signalling in insulin-sensitive target tissues. Ultimately, T2DM and insulin resistance will occur [78,79].

Streptozotocin (STZ) is an antibiotic currently used to reliably induce both insulin-dependent and non-insulin-dependent diabetes mellitus (DM) by inducing beta cell death via DNA alkylation [80]. The STZ-induced DM is a low-cost and fast-onset experimental model applied in many rodent strains [81]. Although high-dose STZ severely impairs insulin secretion, emulating type 1 DM, low-dose STZ has been shown to cause a mild impairment of beta cell insulin secretion similar to late-stage T2DM [82,83]. Thent et al. [84] reported that the aqueous leaf extract of PS (125 mg/kg/day for 28 days) increased body

weight, and decreased fasting blood glucose and urine glucose levels in STZ-induced DM rats. A short-term treatment of the PS aqueous leaf extract (125 and 250 mg/kg/day for 7 days) did not lower the fasting plasma glucose level of STZ-induced diabetic rats [85]. This could be attributed to the short treatment duration, which was insufficient to produce any therapeutic effects. Hussan et al. [86] reported increased inflammatory cells infiltration, Bowman's capsule size, and glomerular membrane thickness in the kidney of STZ-induced DM rats. This suggests that hyperglycaemia due to DM causes kidney damage, leading to renal failure in rats. Some renal complications in DM include diabetic nephropathy, proteinuria, hypertension, and cardiovascular risks [87]. The aqueous leaf extract of PS (125 mg/kg/day for 28 days) reduced these degenerative renal changes in STZ-induced DM rats, suggesting that PS supplementation prevented renal failure due to DM. However, it did not change the body weight, fasting blood glucose, kidney weight index, and glomerular area in a renal corpuscle of DM rats [86].

There were a few shortcomings in the studies examined. Some studies measured fasting blood glucose as an index of the glycaemic status of the animals, which might not be accurate. It should be supported by the oral glucose tolerance test, insulin level, and homeostasis model assessment of insulin resistance, which are stronger predictors of T2DM. At the very least, a combination of fasting plasma glucose and glycated haemoglobin (HbA1c) would be adequate to indicate DM accurately [88]. On the other hand, STZ does not produce hyperinsulinemia associated with early stage T2DM. According to Skovso [89], a combination of a high-fat diet (HFD) and STZ is a better model to recapitulate DM progression in humans. Otherwise, a high-carbohydrate high-fat diet without STZ would also reproduce the hyperinsulinemia state in early DM, but the successful establishment of MetS would take 12–16 weeks [90].

Apart from that, PS alone fails to lower blood glucose levels in two studies [85,86]. Thus, future research should explore combining PS with a conventional T2DM medication such as metformin. Furthermore, only one study examined the effects of PS on renal complications due to DM (diabetic nephropathy). Since DM produces a wide range of complications, future research should also investigate the effects of PS on neuropathy and oculopathy due to T2DM. The anti-inflammatory and antioxidant effects of PS in alleviating T2DM and its complications have not been scrutinised thus far. These processes are important determinants of beta cell survival and function [91]. Figure 3 summarises the current knowledge on the effect of PS supplementation on hyperglycaemia and T2DM and the further investigation required in this area.

### 3.3. Effect of PS on Hypertension

Increased blood pressure is regarded as a critical component of MetS. Even in the absence of diabetes, more than 85% of individuals with MetS have high blood pressure (BP) or hypertension [92]. Oxidative stress and inflammation have been implicated in the pathogenesis of hypertension. Oxidative stress and inflammation increase ROS accumulation, which reduces the bioavailability of the protective vasodilator NO and modulates endothelial function, vascular tone, and cardiac function [93,94]. Furthermore, ROS has been linked to pathological processes, such as inflammation, hypertrophy, apoptosis, fibrosis, and vessel rarefaction. These factors contribute to the development of endothelial dysfunction and cardiovascular remodelling, which are hallmarks of hypertension [95,96]. As a result, reducing oxidative stress, primarily using antioxidant molecules, may be beneficial in preventing and treating hypertension.

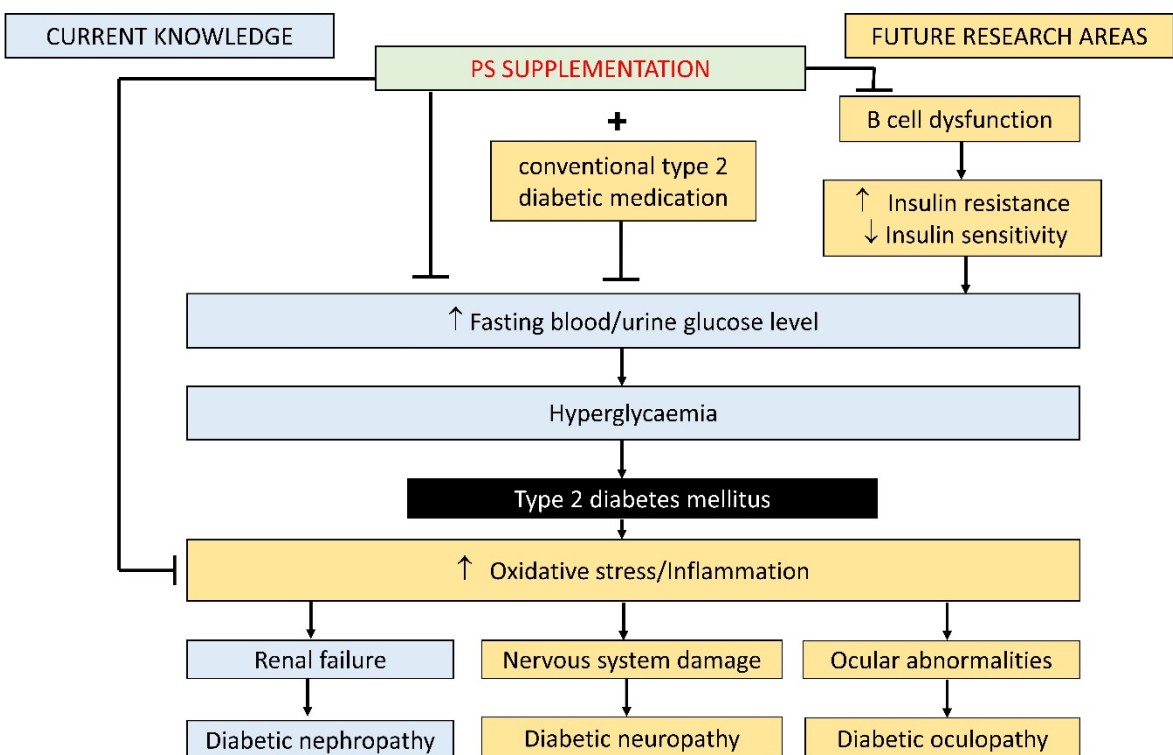

**Figure 3.** Protective effect of PS supplementation against hyperglycaemia and T2DM: current knowledge and future area of investigation. List of abbreviations: PS, *Piper sarmentosum*; ↑, increase; ↓, decrease; +, in combination with.

Several studies reported the hypotensive effect of PS. In male Wistar rats with Nω-nitro-L-arginine methyl ester hydrochloride (L-NAME)-induced hypertension, the aqueous leaf extract of PS (125, 250, and 500 mg/kg/day for 4 weeks) reduced systolic, diastolic, and mean arterial blood pressure [97]. Similar observations were obtained in dexamethasone-induced [98,99] and spontaneously hypertensive rats treated with PS [24,100,101]. The hypotensive effect of PS could be attributed to its ability to increase vasodilation and decrease vasoconstriction. Endothelin-1 (ET-1) is a potent vasoconstrictor that increases blood pressure and contributes to the development of hypertension [102,103]. The aqueous leaf extract of PS (500 mg/kg/day for 28 days) reduced the ET-1 level in the mesenteric artery, suggesting that it reduced vasoconstriction in hypertensive rats. Endothelial nitric oxide synthase (eNOS) is the major enzyme responsible for NO production in the blood vessels. NO is a powerful vasodilator and its reduction can negatively impact endothelial-dependent vasodilation, leading to increased peripheral resistance and blood pressure [104]. The aqueous extract of PS increased the expression and activity of eNOS in the thoracic aorta, increased the serum level of eNOS and increased the levels of NO in the serum and mesenteric artery of L-NAME and dexamethasone-induced and spontaneously hypertensive rats [24,97,98,100,101]. This is suggestive of a vasodilatory effect of PS in hypertensive rats. Asymmetric dimethyl arginine (ADMA) is an endogenous competitive inhibitor of NO synthase, and elevated levels of ADMA inhibits NO synthesis [105]. PS supplementation reduced the plasma ADMA levels in spontaneously hypertensive rats [101], thus preventing the inhibition of NO production in hypertensive rats and helped vasodilation. The circulating MDA level was reduced by PS supplementation in L-NAME-induced hypertensive rats (125, 250, and 500 mg/kg/day of the aqueous leaf extract of PS for 4 weeks) [97] and spontaneously hypertensive rats (SHR) (0.5, 1, and 2 mg/kg/day of aqueous extract of PS for 28 days) [24]. These observations indicate that the antioxidant effects of PS were related to its anti-hypertensive properties. However, supplementation with the aqueous leaf extract of PS did not change the heart rate, lactate dehydrogenase, and creatine phosphokinase in SHRs. These observations suggest that it was ineffective in preventing hypertension-induced cardiac tissue injury [24]. The action of PS could be

nullified because oxidative stress is not involved in the development of hypertension in SHRs. Results from gene expression studies on the brain of SHRs suggest that albumin and chymase in the presence of angiotensinogen and prostaglandin E receptor 4 are responsible for causing hypertension in SHRs [106]. This suggests that the prostaglandin E receptor 4 pathway or the renin-angiotensin-aldosterone system (RAAS) may be responsible for hypertension in SHRs.

The studies that were examined had some limitations. Drug-induced (L-NAME and dexamethasone) and genetic (SHRs) models were used in the studies. SHRs are widely used as a rat model of primary or essential hypertension [107], whereas drug-induced hypertension is mostly used in secondary hypertension studies [108]. Since 90 to 95% of hypertension cases are essential or primary [109], the spontaneous hypertension model may be a better option for future hypertension research. None of the studies examined measured blood vessel contractility directly. This could be performed ex vivo using freshly harvested blood vessels and might provide useful information on the effects of PS on vascular function and contractility, as well as the endothelial-derived mediators in modulating this function [110]. The only clearly illustrated hypertensive mechanism of PS is on the eNOS/NO/ET pathway, and many pathways are yet to be explored. Glucocorticoid-induced inhibition of prostaglandin synthesis and RAAS activation due to renal failure have been reported to cause vasoconstriction and increase water/sodium retention. These factors can increase blood pressure [111–114]. Therefore, future studies should explore the effects of PS on mechanisms such as prostaglandin receptor activation and RAAS. Figure 4 summarises the current knowledge on the effect of PS supplementation on hypertension and the further investigation required in this area.

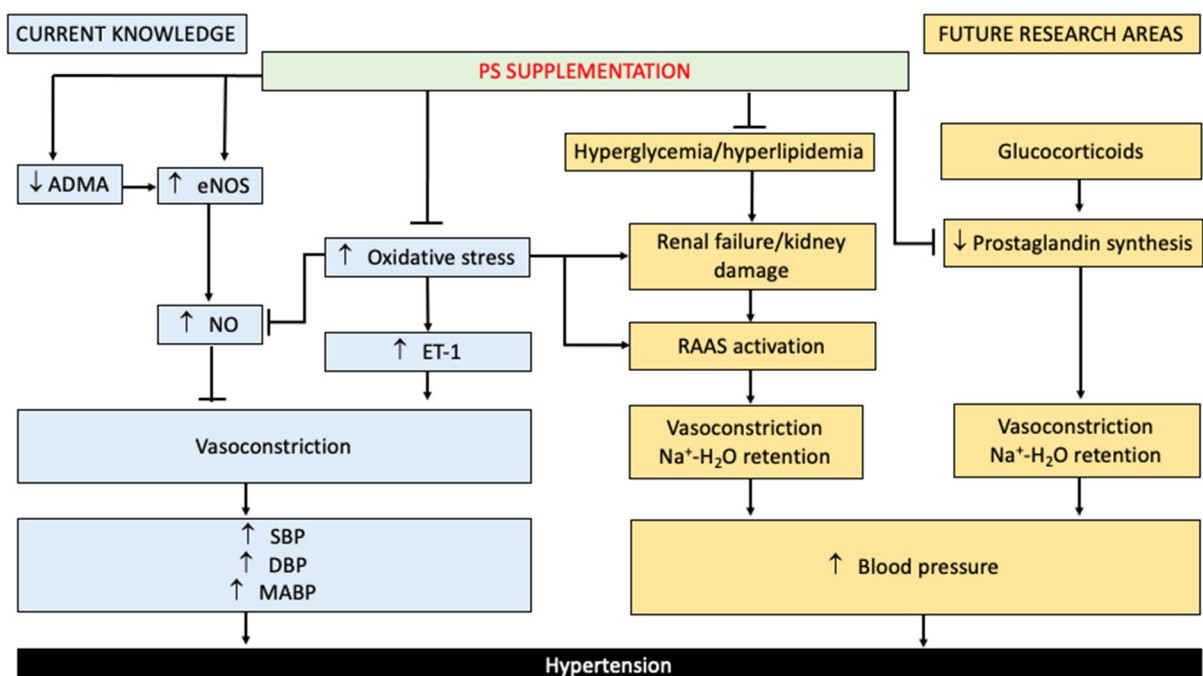

**Figure 4.** Protective effect of PS supplementation against hypertension: current knowledge and future area of investigation. List of abbreviations: PS, *Piper sarmentosum*; ↑, increase; ↓, decrease; ADMA, asymmetric dimethyl arginine; NO, nitric oxide; eNOS, endothelial nitric oxide synthase; SBP, systolic blood pressure; DBP, diastolic blood pressure; MABP, mean arterial blood pressure; ET-1, endothelin 1; RAAS, renin-angiotensin-aldosterone system; $Na^+$, sodium ion; $H_2O$, water.

### 3.4. Effect of PS on Dyslipidaemia and Its Complications

Dyslipidaemia has emerged as a global epidemic. According to the Global Burden of Disease, Injuries, and Risk Factors study, morbidity and mortality attributable to dyslipidaemia worldwide have increased by 26.9 and 28.0%, respectively, over the past few decades [115–117]. Dyslipidaemia is characterised by increased fasting and postprandial

triglyceride-rich lipoproteins, decreased HDL, and increased small, dense low-density lipoprotein (LDL) particles [118,119]. The pathogenesis of dyslipidaemia has been linked to oxidative stress. During the attack of free radicals on membrane lipoproteins and polyunsaturated fatty acids, many oxygenated compounds, particularly aldehydes, such as MDA and conjugated dienes, are produced. Many studies have found that serum MDA levels are higher in hyperlipidaemic subjects and decrease after antioxidant supplementation, with similar findings in animal models of hyperlipidaemia [120,121]. Inflammation is also gaining attention for its potential role in the pathogenesis of dyslipidaemia [122,123], as people with dyslipidaemia generally have higher levels of inflammatory biochemical markers than those without dyslipidaemia [123].

Some studies have reported the hypolipidaemic effect of PS. In rats with fructose-induced MetS, the methanolic leaf extract of PS (125 mg/kg/day for 4 weeks) reduced total cholesterol (TC), triglyceride (TG), and low-density lipoprotein (LDL), and increased high-density lipoproteins (HDL) [68]. Ali et al. [124] also reported similar results in ovariectomy-induced obese rats treated with the aqueous leaf extract of PS (125 mg/kg/day for 5 months). The aqueous leaf extract of PS (500 mg/kg/day for 28 days) also reduced serum TC, TG, and LDL, but did not change the HDL levels [125]. Additionally, 3-hydroxy-3-methyl glutaryl coenzyme A reductase (HMGCR) activity was reduced by PS (the methanolic leaf extract 125 mg/kg/day for 4 weeks) in fructose-induced MetS rats and ovariectomy-induced obese rats (the aqueous leaf extract, 125 mg/kg/day for 5 months). HMGCR is the rate-controlling enzyme of the mevalonate pathway and is responsible for cholesterol synthesis in the liver [126]. A decrease in HMGCR expression/activity indicates a decrease in cholesterol production [127,128]. This suggests that the hypolipidaemic effect of PS is primarily driven by a decrease in HMGCR activity, similar to that of statins. Thus, it could be used as an alternative therapy for hyperlipidaemia [124]. Notably, the methanolic extract of PS (4 weeks) improves the lipid profile in animals faster than the aqueous extract (3–5 months), indicating a better efficacy.

Given the potential effects of PS on HMGCR, it is interesting to examine the effects of statins and PS in combination on the mevalonate pathway. Prolonged statin use has been frequently associated with muscle associated symptoms [129]. Thus, the combination could reduce the dose of statins required to achieve hypocholesterolaemic effects, thus lowering the risk of side effects. However, this speculation awaits validation from future studies. Figure 5 summarises the current knowledge on the effect of PS supplementation on hyperlipidaemia and the further investigation required in this area.

Overall, PS supplementation attenuated the adverse effects associated with MetS. In vivo studies showed that PS reduced the visceral fat deposition and diameter of adipocytes as well as increasing the adiponectin levels, thereby reducing the visceral fat accumulation and lipid content in adipocytes in animals. PS also increased vasodilation and decreased vasoconstriction, thereby decreasing SBP, DBP, and MABP in animals. PS supplementation reduced the fasting blood/urine glucose levels and prevented renal failure in diabetic rats. PS also reduced HMGCR expression, TG, TC, and LDL, and increased HDL in hyperlipidaemic rats. An overview of the effects of PS on components of MetS is presented in Table 1. The literature search revealed a lack of in vitro and clinical trials validating the effect of PS on MetS and its components. These research gaps should be considered in future studies.

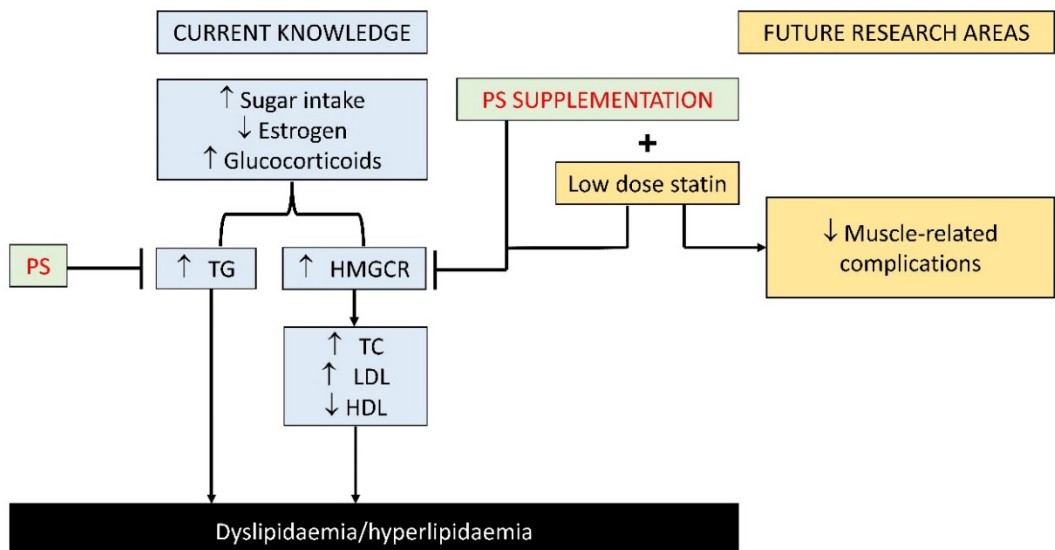

**Figure 5.** Protective effect of PS supplementation against dyslipidaemia/hyperlipidaemia: current knowledge and future area of investigation. List of abbreviation: PS, *Piper sarmentosum*; ↑, increase; ↓, decrease; +, in combination; TG, triglyceride; TC, total cholesterol; LDL, low-density lipoprotein; HDL, high-density lipoprotein; HMGCR, 3-hydroxy-3-methyl glutaryl coenzyme A reductase.

**Table 1.** Protective effect of PS against MetS components.

| Researcher | Study Design | Findings |
|---|---|---|
| | Obesity | |
| Azlina et al. [53] | Animals: 42 female Sprague Dawley rats (180–200 g)<br>Mode of disease induction: ovariectomy-induced obesity<br>Treatment: 125 mg/kg/day of AEPS for 5 months<br>Control:<br>Negative: no treatment<br>Positive: 120 mg/kg/day of GCA for 5 months | ↓ 11βHSD-1 activity in liver and adipose tissue compared to negative control<br>↔ 11βHSD-1 activity in heart compared to negative control<br>↔ blood pressure at 3 and 5 months compared to negative control<br>↔ body weight compared to negative control |
| Azlina et al. [67] | Animals: 28 female Sprague Dawley rats (180–200 g)<br>Mode of disease induction: ovariectomy-induced obesity<br>Treatment: 125 mg/kg/day of AEPS for 5 months<br>Control:<br>Negative: no treatment<br>Positive: 120 mg/kg/day of GCA for 5 months | ↓ blood glucose level at 3 and 5 months compared to negative control<br>↑ plasma adiponectin level at 3 and 5 months compared to negative control<br>↔ body weight compared to negative and positive control |
| Kumar et al. [68] | Animals: 40 male Wistar rats (180–200 g)<br>Mode of disease induction: fructose-induced MetS<br>Treatment: 125 mg/kg/day of MEPS for 4 weeks<br>Control:<br>Negative: no treatment<br>Positive: 100 mg/kg/day of naringin for 4 weeks | ↓ food, fluid, and calorie intake compared to negative control<br>↓ body weight compared to negative control at week 2; negative and positive control at week 4<br>↓ fat mass and fat percentage compared to negative and positive control<br>↓ serum leptin and adiponectin compared to negative and positive control<br>↓ adipocyte surface area compared to negative control |
| Fairus et al. [47] | Animals: 21 male Sprague Dawley rats (200–250 g)<br>Mode of disease induction: adrenalectomy + dexamethasone-induced visceral obesity<br>Treatment: 125 mg/kg/day of PS extract for 48 days<br>Control:<br>Negative: no treatment<br>Positive: 240 mg/kg/day of GCA for 48 days | ↓ visceral fat deposition compared to negative control<br>↓ diameter of adipocyte membrane compared to positive and negative control<br>↔ area, perimeter, and width of individual perirenal adipocytes compared to positive and negative control |

**Table 1.** *Cont.*

| Researcher | Study Design | Findings |
|---|---|---|
| | Diabetes | |
| Thent et al. [84] | Animals: 24 male Sprague Dawley rats (200 ± 50 g)<br>Mode of disease induction: STZ-induced diabetes<br>Treatment: 125 mg/kg/day of AEPS for 28 days<br>Control:<br>Negative: no treatment<br>Positive: no | ↑ body weight and ↓ fasting blood glucose and urine glucose level compared to negative control |
| Hussan et al. [86] | Animals: 18 male Sprague Dawley rats (150 ± 50 g)<br>Mode of disease induction: STZ-induced diabetes<br>Treatment: 125 mg/kg/day of AEPS for 28 days<br>Control:<br>Negative: no treatment<br>Positive: no | ↔ body weight, fasting blood glucose, kidney weight index, and percent glomerular area in a renal corpuscle compared to negative control<br>↓ inflammatory cells infiltration, size of urinary space, and glomerular membrane thickening in kidney compared to negative control. |
| Peungvicha et al. [85] | Animals: 18 male Wistar rats (5 weeks old; 140–220 g)<br>Mode of disease induction: STZ-induced diabetes<br>Treatment: 125 and 250 mg/kg/day of AEPS for 7 days<br>Control:<br>Negative: no treatment<br>Positive: glibenclamide (5 mg/kg/day) for 7 days | ↔ fasting plasma glucose level compared to positive and negative control |
| | Hypertension | |
| Alwi et al. [97] | Animals: 36 adult male Wistar rats (6–8 weeks old; 170–220 g)<br>Mode of disease induction: L-NAME-induced hypertension<br>Treatment: 125, 250, and 500 mg/kg/day of AEPS for 4 weeks<br>Control:<br>Negative: no treatment<br>Positive: no | ↓ SBP and MABP at week 2 and 4 compared to negative control<br>↓ DBP in 250 and 500 mg/kg at week 2 and all concentrations at week 4 compared to the negative control.<br>↓ serum MDA compared to negative control<br>↑ serum NO compared to negative control |
| Azmi et al. [99] | Animals: 30 adult male Sprague Dawley rats (8–12 weeks old; 250–300 g)<br>Mode of disease induction: dexamethasone-induced hypertension<br>Treatment: 500 mg/kg/day of AEPS for 28 days<br>Control:<br>Negative: no treatment<br>Positive: captopril (40 mg/kg/day) for 28 days | ↓ SBP, DBP, and MABP at day 14 and 28 compared to negative control |
| Fadze et al. [98] | Animals: 30 male Sprague Dawley rats (8–12 weeks old)<br>Mode of disease induction: dexamethasone-induced hypertension<br>Treatment: 500 mg/kg/day of AEPS for 28 days<br>Control:<br>Negative: no treatment<br>Positive: captopril (40 mg/kg/day) for 28 days | ↓ SBP, DBP, and MABP at day 14 and 28 compared to negative control<br>↑ eNOS expression in thoracic aorta compared to negative control<br>↑ eNOS protein level in thoracic aorta tissues compared to negative control<br>↑ eNOS activity in thoracic aorta compared to negative control.<br>↑ serum eNOS compared to negative control |
| Fauzy et al. [100] | Animals: 24 male spontaneously hypertensive rats (8–12 weeks old; 250–300 g)<br>Mode of disease induction: spontaneous hypertension<br>Treatment: 500 mg/kg/day of AEPS for 28 days<br>Control:<br>Negative: no treatment<br>Positive: perindopril (3 mg/kg/day) for 28 days | ↓ SBP, DBP, and MABP compared to negative control<br>↔ HR compared to negative and positive control<br>↓ ET-1 and ↑ NO in mesenteric artery compared to negative control |

**Table 1.** *Cont.*

| Researcher | Study Design | Findings |
|---|---|---|
| *Hypertension* | | |
| Mohd Zainudin et al. [101] | Animals: 24 male spontaneously hypertensive rats (8–12 weeks old; 250–300 g)<br>Mode of disease induction: spontaneous hypertension<br>Treatment: 500 mg/kg/day of AEPS for 28 days<br>Control:<br>Negative: no treatment<br>Positive: perindopril (3 mg/kg/day) for 28 days | ↓ SBP and DBP compared to negative control<br>↑ serum NO levels compared to negative control<br>↓ plasma ADMA levels compared to negative control<br>↔ plasma arginine levels compared to negative control |
| Zainudin et al. [24] | Animals: 32 male spontaneously hypertensive rats (10 weeks old)<br>Mode of disease induction: spontaneous hypertension<br>Treatment: 0.5, 1, and 2 mg/kg/day of AEPS for 28 days<br>Control:<br>Negative: Normotensive male Wistar rats (250 ± 10 g)<br>Positive: no treatment | ↓ SBP, DBP, and MABP from week 2–5 compared to positive control<br>↔ HR, CPK, and LDH compared to positive control<br>↑ NO and ↓ MDA in serum compared to positive control<br>↓ serum cholesterol at 1 mg/kg compared to positive control |
| *Dyslipidaemia* | | |
| Kumar et al. [68] | Animals: 40 male Wistar rats (180–200 g)<br>Mode of disease induction: fructose-induced MetS<br>Treatment: 125 mg/kg/day of MEPS for 4 weeks<br>Control:<br>Negative: no treatment<br>Positive: 100 mg/kg/day of naringin for 4 weeks | ↓ Plasma LDL, TC, TG, and HMGCR compared to positive and negative control<br>↓ HMGCR enzyme bioactivity compared to positive and negative control<br>↑ Plasma HDL compared to positive and negative control |
| Ali et al. [124] | Animals: 40 female Sprague Dawley rats (180–200 g)<br>Mode of disease induction: ovariectomy-induced obesity<br>Treatment: 125 mg/kg/day of AEPS for 3 and 5 months<br>Control:<br>Negative: no treatment<br>Positive: 120 mg/kg/day of GCA for 3 and 5 months | ↓ Plasma LDL, TC, TG, and HMGCR at 3 and 5 months compared to negative control<br>↓ HMGCR enzyme bioactivity at 3 and 5 months compared to negative control<br>↑ Plasma HDL at 3 and 5 months compared to negative control |
| Fadze et al. [125] | Animals: 30 male Sprague Dawley rats (180–200 g)<br>Mode of disease induction: dexamethasone-induced hyperlipidaemia<br>Treatment: 500 mg/kg/day of AEPS for 28 days<br>Control:<br>Negative: no treatment<br>Positive: 40 mg/kg/day of captopril for 28 days | ↓ LDL, TC, and TG compared to negative control<br>↔ HDL compared to negative control |

Abbreviations: ↑, increase; ↓, decrease; ↔, no change; L-NAME, Nω-nitro-L-arginine methyl ester hydrochloride; DBP, diastolic blood pressure; SBP, systolic blood pressure; MDA, malondialdehyde; MABP, mean arterial blood pressure; NO, nitric oxide; AEPS, aqueous extract of *Piper sarmentosum*; PS, *Piper sarmentosum*; MEPS, methanolic extract of *Piper sarmentosum*; ET-1, endothelin-1; FBS, fasting blood serum; HR, heart rate; CPK, creatinine phosphokinase; LDH, lactate dehydrogenase; ADMA, asymmetric dimethyl arginine; STZ, streptozocin; eNOS, endothelial nitric oxide synthase; 11βHSD-1, 11β-Hydroxysteroid dehydrogenase type 1; GCA, glycyrrhizic acid; HDL, high-density lipoprotein; LDL, low-density lipoprotein; TG, triglyceride; TC, total cholesterol; HMGCR, 3-hydroxy-3-methyl-glutaryl-coenzyme A reductase.

## 4. Effect of PS on Cardiovascular Diseases Linked to MetS

Cardiovascular diseases contribute significantly to morbidity and mortality due to non-communicable diseases globally [130]. The risks of atherosclerosis and cardiovascular deaths are increased in populations with MetS due to their proinflammatory and pro-thrombotic state [131]. Chronic non-resolving systemic inflammation and oxidative stress due to MetS were associated with cardiovascular disease severity and complex vascular lesions that are prone to rupture, resulting in acute complications [132].

The protective effects of PS on cardiovascular diseases have been reported in several in vitro studies. In a study by Ismail et al. [133], TNF-α-induced HUVECs treated with the aqueous leaf extract of PS (100, 150, 250, and 300 µg/mL for 24 h) showed a reduction in vascular cell adhesion molecule-1 (VCAM-1) and intercellular adhesion molecule-1 (ICAM-1) protein expression [118]. These are adhesion cell molecules expressed by endothelial cells

due to endothelium damage in the early stages of atherosclerosis development [134]. This effect was achieved by inhibiting the nuclear factor-kappa B (NF-κB) pathway, evidenced by a reduction in TNF-α-stimulated NF-κB p65 subunit expression [133]. TNF-α-stimulated HUVECs supplemented with the aqueous leaf extract of PS (150, 250, and 300 μg/mL for 24 h) showed decreased ADMA levels, increased NO levels, and an increased expression of dimethylarginine dimethylaminohydrolase 1 (DDAH 1) mRNA and protein [135]. Similarly, $H_2O_2$-stimulated HUVECs supplemented with the aqueous leaf extract of PS (150 μg/mL for 24 h) showed an increased mRNA and protein expression and enzymes activity of eNOS and an increased production of NO [136]. Dimethylarginine dimethylaminohydrolase (DDAH) is an enzyme that degrades ADMA [137], and a decrease in its activity leads to an increase in ADMA, which reduces eNOS activity and NO production [138]. A reduced nitric oxide synthesis due to eNOS downregulation plays a significant role in vasoactive factor imbalance, endothelial haemostasis impairment, and the early development of atherosclerosis [139]. These studies suggest that PS could reduce the risk of endothelial dysfunction and atherosclerosis due to its anti-inflammatory properties through the inhibition of the NF-κB pathway and stimulation of the DDAH/NOS pathway.

An increase in inflammatory markers is linked to hypercholesterolaemic atherosclerosis. In animal studies, high cholesterol diet (HCD)-induced atherosclerotic rabbits treated with the aqueous leaf extract of PS (500 mg/kg/day for 10 weeks) showed decreased ICAM, VCAM, and C-reactive protein (CRP) [140]. CRP is important in the pathogenesis of the vascular inflammatory process. It promotes endothelial cells to express ICAM-1 and VCAM-1 [141], suggesting that PS may help to reduce the inflammation linked to the development of atherosclerosis due to hypercholesterolaemia. The aqueous leaf extract (500 mg/kg/day for 10 weeks) decreased the fat deposit and atherosclerotic lesion in the aortic intima layer of atherogenic rabbits [142]. The PS aqueous leaf extract (250 and 500 mg/kg/day for 10 weeks) [142] and the methanolic leaf extract (62.5, 125, and 250 mg/kg/day for 10 weeks) [22] reduced the thickening of the intima ratio, neointimal area, fatty streak, and foam cells covering the intima layer in the abdominal aorta of atherogenic rabbits. This implies that PS supplementation could reduce the blood vessel narrowing associated with atherosclerosis by preventing fat deposition and reducing atherosclerotic lesions in hypercholesterolaemic rabbits. Rabbit is a better model of atherosclerosis because rats are resistant to atherogenesis [143], even at a 10% *w/w* cholesterol diet [143]. This is because lipid metabolism in rats is primarily based on high-density lipoprotein (HDL), rather than LDL, as is the case in humans [144]. As a result, additional agents such as vitamin D3 are frequently required to produce atherosclerotic calcification or aortic calcification in rats [145]. On the other hand, rabbits and hamsters are sensitive to dietary cholesterol-induced atherosclerosis [146]. In particular, hamsters carry a significant portion of their plasma cholesterol in LDL particles, making them similar to humans [147] and sensitive to high-fat diets [148].

Several studies have implicated DM in the development of cardiac dysfunction and atherosclerosis as metabolic deterioration in DM leads to structural and biochemical changes in the heart [84]. STZ-induced diabetic rats treated with the aqueous leaf extract of PS (125 mg/kg/day for 28 days) showed reduced disturbance and irregular arrays of myofibrils within the sarcomere of cardiac tissues. The size, disruption, and patchy areas of cytoplasmic space in the mitochondria of cardiac tissue were also reduced [84]. Similarly, Thent et al. [149] reported a decreased deformation in the sizes and shapes of the cardiomyocyte's nuclei of cardiac tissue. They also reported a decreased thickness of the tunica media of aorta and decreased disruption in the arrangement of the elastic fibres in aortic tissue diabetic rats. PS leaf extract could prevent cardiac dysfunction and early signs of atherosclerosis by reducing degenerative changes in the myocardium and aortic tissues of STZ diabetic rats.

There are a couple of drawbacks worth considering. All the in vitro studies did not use well-established antioxidant or anti-inflammatory agents as a positive control to verify the action of PS. Although simvastatin did not lower CRP, it did reduce VCAM and ICAM

more effectively than PS, according to a study by Amran et al. [140]. The combination of PS and simvastatin may be more effective in preventing atherosclerosis than individual agents. Table 2 summarises the effect of PS on atherosclerosis linked to MetS, while Figure 6 summarises the current knowledge on the effect of PS supplementation on cardiovascular diseases linked to MetS and the further investigation required in this area.

**Table 2.** Effect of PS on atherosclerosis and cardiovascular complications due to MetS.

| Researcher | Study Design | Findings |
|---|---|---|
| | Cell Culture Studies | |
| Ismail et al. [133] | Cell line: HUVECs<br>Mode of disease induction: TNF-$\alpha$-induced atherosclerosis<br>Treatment: 100, 150, 250, and 300 µg/mL AEPS for 24 h<br>Control:<br>Negative: no treatment<br>Positive: no | $\downarrow$ ICAM-1 and NF-$\kappa$B p65 protein expression at 150, 250, and 300 µg/mL compared to negative control<br>$\downarrow$ VCAM-1 protein expression compared to negative control |
| Ugusman et al. [136] | Cell line: HUVECs<br>Mode of disease induction: $H_2O_2$-induced atherosclerosis<br>Treatment: 150 µg/mL AEPS for 24 h<br>Control:<br>Negative: no treatment<br>Positive: no | $\uparrow$ expression of eNOS mRNA compared to negative control<br>$\uparrow$ eNOS protein levels compared to negative control<br>$\uparrow$ eNOS enzyme activity compared to negative control<br>$\uparrow$ NO production compared to negative control<br>$\uparrow$ NO levels compared to negative control |
| Sundar et al. [135] | Cell line: HUVECs<br>Mode of disease induction: TNF-$\alpha$-induced atherosclerosis<br>Treatment: 150, 250, and 300 µg/mL AEPS for 24 h<br>Control:<br>Negative: no treatment<br>Positive: no | $\uparrow$ DDAH1 mRNA expression, protein level, and enzyme activity compared to negative control<br>$\downarrow$ ADMA level compared to negative control |
| | Animal Studies | |
| Amran et al. [140] | Animals: 42 male New Zealand White rabbits (1.8 $\pm$ 2 kg)<br>Mode of disease induction: HCD-induced atherosclerosis<br>Treatment: 62.5, 125, 250, and 500 mg/kg/day of AEPS for 10 weeks<br>Control:<br>Negative: no treatment<br>Positive: simvastatin (1.2 mg/kg) for 10 weeks | $\downarrow$ ICAM, VCAM, and CRP at 500 mg/kg compared to negative control |
| Amran et al. [142] | Animals: 42 male New Zealand White rabbits (1.8 $\pm$ 2 kg)<br>Mode of disease induction: HCD-induced atherosclerosis<br>Treatment: 62.5, 125, 250, and 500 mg/kg/day of AEPS for 10 weeks<br>Control:<br>Negative: no treatment<br>Positive: simvastatin (1.2 mg/kg) for 10 weeks | $\downarrow$ atherosclerotic lesions and fat deposit in the intimal surface of the aorta at 500 mg/kg compared to negative control<br>$\downarrow$ thickening of intimal ratio in the abdominal aorta at 250 and 500 mg/kg compared to negative control<br>$\downarrow$ foam cells in intima layer of the abdominal aorta at 250 and 500 mg/kg compared to negative control |

<div align="center">

**Table 2.** *Cont.*

</div>

| Researcher | Study Design | Findings |
|---|---|---|
| | Animal Studies | |
| Amran et al. [22] | Animals: 36 male New Zealand White rabbits (1.8 ± 2 kg)<br>Mode of disease induction: HCD-induced atherosclerosis<br>Treatment: 62.5, 125, and 250 mg/kg/day of MEPS for 10 weeks<br>Control:<br>Negative: no treatment<br>Positive: simvastatin (1.2 mg/kg) for 10 weeks | ↓ fatty streak in abdominal aorta compared to negative control<br>↓ neointimal area and intima ratio of abdominal aorta compared to negative control<br>↓ foam cells covering intima layer of abdominal aorta compared to negative control |
| Thent et al. [149] | Animals: 24 male Sprague Dawley rats (200–250 g)<br>Mode of disease induction: STZ-induced diabetes<br>Treatment: 125 mg/kg/day of AEPS for 28 days<br>Control:<br>Negative: no treatment<br>Positive: no | ↓ deformation in sizes and shapes of the cardiomyocytes nuclei of cardiac tissue compared to negative control<br>↓ thickness of the tunica media of aortic wall compared to negative control<br>↓ disruption in the arrangement of the elastic fibres in aortic tissue compared to negative control |
| Thent et al. [84] | Animals: 24 male Sprague Dawley rats (200 ± 50 g)<br>Mode of disease induction: STZ-induced diabetes<br>Treatment: 125 mg/kg/day of AEPS for 28 days<br>Control:<br>Negative: no treatment<br>Positive: no | ↓ disturbance and irregular arrays of myofibrils within sarcomere of cardiac tissues compared to negative control<br>↓ size, disruption, and patchy areas of cytoplasmic space in mitochondria of cardiac tissue compared to negative control<br>↓ invagination and disruption of nuclei in cardiac tissue compared to negative control<br>↓ disruption of the elastic lamina, proliferation of smooth muscle cells and ↑ endothelial cells in proximal aorta compared to negative control |

Abbreviations: ↑, increase; ↓, decrease; HUVECs, human umbilical vein endothelial cells; VCAM-1, vascular cell adhesion molecule-1; ICAM-1, intercellular adhesion molecule-1; TNF-α, tumour necrosis factor-α; FBS, fasting blood glucose; CRP, c-reactive protein; STZ, streptozocin; NO, nitric oxide; ADMA, asymmetric dimethyl arginine; DDAH1, dimethylarginine dimethylaminohydrolase 1; HCD, high cholesterol diet; AEPS, aqueous extract of *Piper sarmentosum*; NF-κB, nuclear factor-kappa B.

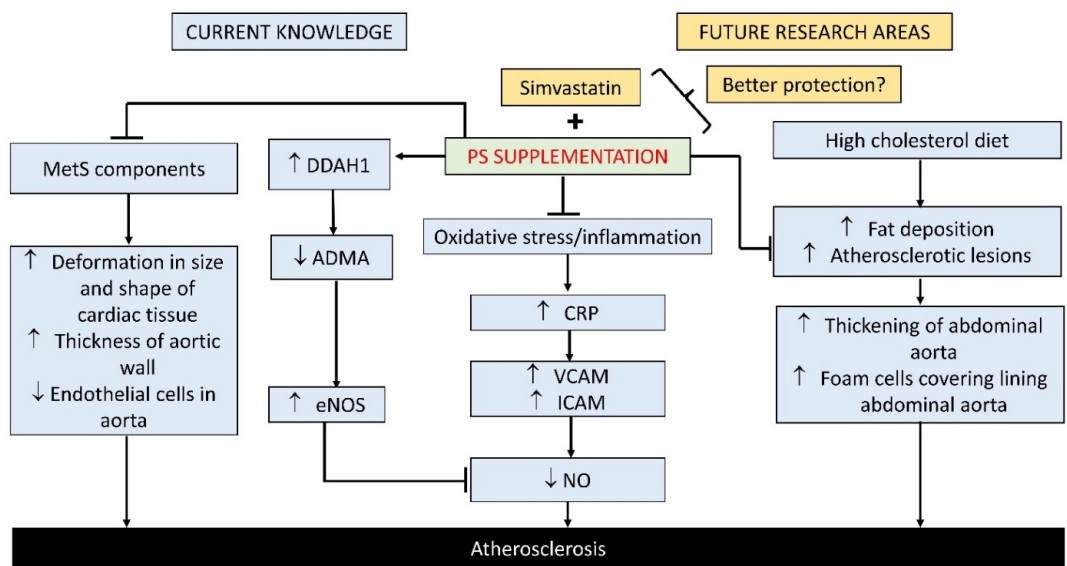

**Figure 6.** Protective effect of PS supplementation against dyslipidaemia/hyperlipidaemia: current knowledge and future area of investigation. List of abbreviation: PS, *Piper sarmentosum*; ↑, increase; ↓, decrease; +, in combination; DDAH-1, dimethylarginine dimethylaminohydrolase 1; ADMA, asymmetric dimethyl arginine; NO, nitric oxide; eNOS, endothelial nitric oxide synthase; CRP, C-reactive protein; VCAM, vascular cell adhesion molecule-1; ICAM, intercellular adhesion molecule-1.

### 5. Safety and Bioavailability of PS

Despite the medical properties of PS, information on its toxicity and bioavailability is limited. Peungvicha et al. [85] found that the $LD_{50}$ of the aqueous whole plant extract of PS was more than 10 g/kg through the oral route in rats [85], while that of the methanolic leaf extract of PS was more than 5 g/kg in mice [150]. The aqueous leaf extract of PS did not cause subacute toxicity in Sprague Dawley rats at 2 g/kg based on their haematological profile and liver and kidney histology [151]. Similarly, Hussian et al. [152] reported that the $LD_{50}$ of ethanolic extracts of PS fruit and leaves was more than 2 g/kg, and no adverse behavioural change, signs of toxicity, and mortality were observed [152].

PS has been found to contain several biologically active amides, including pellitorine, sarmentine, and sarmentosine. According to Hussain et al. [153], pellitorine was found in the intestinal wall, liver, lungs, kidney, and heart after an oral administration of the ethanolic extract of PS fruit at a dose of 500 mg/kg, whereas sarmentine was only found in the intestinal wall and heart of male Sprague Dawley rats. Sarmentosine, on the other hand, was not detected in plasma, tissues, or urine, but was found in faeces. According to their findings, pellitorine and sarmentine, two bioactive markers of PS, have good oral bioavailability and different tissue affinities, and are excreted in urine as metabolites, whereas sarmentosine has a low oral bioavailability and is not absorbed in the intestine [153]. These findings will aid in interpreting the efficacy of PS in preclinical studies and predicting human pharmacokinetics using the scaling technique. A search on https://clinicaltrials.gov/ (accessed on 30 August 2021) using the term "*Piper sarmentosum*" yielded no results as there were no clinical trials available for PS. Hence, the effects of PS in humans cannot be validated.

### 6. Conclusions

Based on the literature, PS may have a therapeutic effect against MetS, and cardiovascular disease linked to MetS. The therapeutic effects of PS on MetS may be due to its ability to reduce inflammation and oxidative stress, making PS a possible therapeutic or adjuvant treatment for MetS and associated complications in future. However, the effects of PS on hyperglycaemia are inconsistent and confounded by the use of fasting blood glucose as the sole indicator of glycaemic status without further comprehensive assays. Despite the generally positive results from preclinical studies, there is no human clinical trial to validate these claims. As a result, well-designed human clinical trials should be carried out to validate the beneficial health effects of PS.

**Author Contributions:** Conception, S.O.E., M.F.N.A. and K.-Y.C.; literature search and data extraction, S.O.E. and K.-Y.C.; writing—original draft preparation, S.O.E.; writing—review and editing, M.F.N.A. and K.-Y.C.; project administration and funding, K.-Y.C. All authors have read and agreed to the published version of the manuscript.

**Funding:** This research was funded by Universiti Kebangsaan Malaysia through Research University Grant (GUP-2020-021). Sophia Ogechi Ekeuku is a postdoctoral researcher funded by Universiti Kebangsaan Malaysia through FPR-1.

**Institutional Review Board Statement:** Not applicable.

**Informed Consent Statement:** Not applicable.

**Data Availability Statement:** Not applicable.

**Conflicts of Interest:** The authors report no conflicts of interest in this work.

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
