# Peer review of "Effects of Piper sarmentosum on Metabolic Syndrome and Its Related Complications: A Review of Preclinical Evidence"

_applsci, doi:10.3390/app11219860_

Round 1
Reviewer 1 Report
Reviewer comments and suggestions
This review updated the available literature on the effects of Piper sarmentosum (PS) on different criteria of MetS and their complications. The study discussed the useful properties of PS on various metabolic risk factors such as adiposity, hyperglycaemia, hypertension, and dyslipidemia. Due to the antioxidant and anti-inflammatory properties of (PS), it protected various MetS-associated cardiovascular complications. However, the authors suggested that it is prudent to use it in randomised controlled trials in humans so that its efficacy could be explored completely.
Below are the suggested comments to be incorporated in the revised version of the manuscript.
- Line 30-32 needs appropriate references.
- Line 40-41 The lines need to be explored with the help of references from other studies
- Line 120-121 need references
- What do the sentences indicate “Weight gain and increased visceral adiposity are both associated with chronic glucocorticoid therapy”
- Line 130-132 The sentence need to modify
- Line 137-138 Better to present some introductory lines on leptin
- Line 143 It should be better discussed on upper para
- Line 146 better to discussed a few study based on sex differences
- Line 163-165 incomplete discussion of the paper
- Line 168-169 is there was any study please cite leptin resistance n PS
- Line 177 is this study reported some adipokines level
- Please avoid long sentence 210-214, 271-276 315-319 381-391 415-421and 434-439 etc. , please check in the MS
- The nervous system and oculopathy did not discuss in the MS shown in Figure 2.
- The FIGURE 3 pictures need to be describe in all IN THE TEXT
- In many portion, the authors written many studies but cited only one, its not appropriate to use this term if you are using several
- What do the authors want to indicate line 422-423
- In conclusion Not a good word of choice here “MetS-suppressing effects”
- I have seen the authors used many references that review, it would be better to cite research articles related to METS and antioxidants. I have coded some, please try to include the research manuscript.
Prevalence of metabolic syndrome in type 2 diabetes mellitus patients (nih.gov)
Prevalence of metabolic syndrome in type 2 diabetes mellitus using NCEP-ATPIII, IDF and WHO definition and its agreement in Gwalior Chambal region of Central India - PubMed (nih.gov)
Metabolic syndrome as a predictor of type 2 diabetes, and its clinical interpretations and usefulness (nih.gov)
Associations between Dietary Antioxidant Intake and Metabolic Syndrome (plos.org)
Association of antioxidant status and inflammatory markers with metabolic syndrome in Thais | Journal of Health, Population and Nutrition | Full Text (biomedcentral.com)
Status of antioxidant and lipid peroxidation in type 2 diabetic human subjects diagnosed with and without metabolic syndrome by using NCEP-ATPIII, IDF and WHO criteria - ScienceDirect
The Metabolic Syndrome and Antioxidant Concentrations | Diabetes (diabetesjournals.org)
Author Response
Dear reviewer,
Thank you for reviewing our manuscript. We appreciate the constructive comments provided and have responded to each of them in the attached response sheet. Changes in the text are marked with yellow highlights.
Thank you once again for taking the time to examine our response. We are hopeful that the revised manuscript will meet the journal's standards.

Reviewer 2 Report
The manuscript submitted to Applied Sciences by Ekeuku et al., titled: “Effects of Piper sarmentosum on Metabolic Syndrome and its Related Complications”, is a review discussing evidence regarding the effects of a plant (Piper sarmentosum) on Metabolic Syndrome.
The concept of the manuscript is interesting but there seems to be somewhat of an overbidding for the functions of Piper sarmentosum. The evidence provided do not seem to come from RCT which while the authors recognize and mention it as a limitation it becomes a serious drawback as per the confidence in the data and conclusions derived. Furthermore a lot of the information provided is irrelevant with the topic (PS effects on metabolic syndrome). For example common knowledge when describing diabetes and pharmacological approaches to diabetes should be omitted since the focus of the manuscript is arguably different. Keep it focused, clean and neat otherwise it becomes too cumbersome and challenging to follow since the focus is not clear. The reviewer would suggest shortening and focusing the manuscript more.
The reviewer would like to offer the following points for improvement:
- The manuscript needs proofreading for spelling, grammar and syntax improvements.
- The studies referenced in the section of PS on glycaemia are mostly looking at FPG. The authors mention that this is a serious weakness yet they include the studies in the discussion. FPG is an inadequate measurement for the determination of T2DM or glycaemia control. The worth of including those studies and their results is questionable given the fact that they may be misleading in terms of forming conclusions.
- One of the limitations of the manuscript is that it is very weak in reporting human work derived data. One cannot claim effects translatable to humans without proper studies involving human participants. It is important for the title to be modified accordingly.
Author Response

(The authors gave the same response as above.)

Round 2
Reviewer 2 Report
The authors made a reasonable effort to address reviewer's comments.
Author Response
Thank you for the reply. There is no comment to be addressed.
This manuscript is a resubmission of an earlier submission. The following is a list of the peer review reports and author responses from that submission.